# HMGB1 Promotes In Vitro and In Vivo Skeletal Muscle Atrophy through an IL-18-Dependent Mechanism

**DOI:** 10.3390/cells11233936

**Published:** 2022-12-06

**Authors:** Trung-Loc Ho, Chih-Hsin Tang, Sunny Li-Yun Chang, Chun-Hao Tsai, Hsien-Te Chen, Chen-Ming Su

**Affiliations:** 1Graduate Institute of Biomedical Sciences, College of Medicine, China Medical University, Taichung City 40402, Taiwan; 2Department of Pharmacology, School of Medicine, China Medical University, Taichung City 40402, Taiwan; 3Chinese Medicine Research Center, China Medical University, Taichung City 40402, Taiwan; 4Department of Biotechnology, College of Health Science, Asia University, Taichung City 41354, Taiwan; 5Department of Sports Medicine, China Medical University, Taichung City 40402, Taiwan; 6Department of Orthopedic Surgery, China Medical University Hospital, Taichung City 40447, Taiwan; 7School of Medicine, China Medical University, Taichung City 40402, Taiwan; 8Spine Center, China Medical University Hospital, China Medical University, Taichung City 40447, Taiwan

**Keywords:** HMGB1, IL-18, inflammation, myogenesis, skeletal muscle atrophy

## Abstract

Skeletal muscle atrophy occurs due to muscle wasting or reductions in protein associated with aging, injury, and inflammatory processes. High-mobility group box-1 (HMGB1) protein is passively released from necrotic cells and actively secreted by inflammatory cells, and is implicated in the pathogenesis of various inflammatory and immune diseases. HMGB1 is upregulated in muscle inflammation, and circulating levels of the proinflammatory cytokine interleukin-18 (IL-18) are upregulated in patients with sarcopenia, a muscle-wasting disease. We examined whether an association exists between HMGB1 and IL-18 signaling in skeletal muscle atrophy. HMGB1-induced increases of IL-18 levels enhanced the expression of muscle atrophy markers and inhibited myogenic marker expression in C2C12 and G7 myoblast cell lines. HMGB1-induced increases of IL-18 production in C2C12 cells involved the RAGE/p85/Akt/mTOR/c-Jun signaling pathway. HMGB1 short hairpin RNA (shRNA) treatment rescued the expression of muscle-specific differentiation markers in murine C2C12 myotubes and in mice with glycerol-induced muscle atrophy. HMGB1 and IL-18 signaling was suppressed in the mice after HMGB1 shRNA treatment. These findings suggest that the HMGB1/IL-18 axis is worth targeting for the treatment of skeletal muscle atrophy.

## 1. Introduction

Skeletal muscle accounts for around 40% of total body mass in humans and is essential for performing physical functions. Skeletal muscle atrophy is an adverse physiological consequence of muscle wasting due to injury, denervation, inflammation, or aging [1,2]. Muscle atrophy reduces the quality of life and increases both morbidity and mortality [3]. Although several molecular mechanisms underlying muscle loss have been explored [4], no pharmacological therapy exists for skeletal muscle atrophy [5].

Muscle atrophy derives from musculoskeletal disuse following muscle injury [6] Muscle injury-related disuse is associated with the changes in muscle protein synthesis and degradation [7]. Molecular investigations into protein degradation related to skeletal muscle loss have focused on muscle RING-finger-1 (MuRF-1) and MuRF-2 as markers of skeletal muscle atrophy [8], on the ubiquitin ligase atrogin-1 [9], and on myostatin as a biomarker of muscle wasting [10]. The myogenic markers myogenic differentiation (MyoD), paired box 7 (Pax-7), and insulin-like growth factor 1 (IGF-1) promote muscle protein synthesis and maintain muscular function [11,12]. In addition, an abundance of inflammatory cytokines in response to muscle injury may promote alterations of muscle atrophy markers and myogenic markers [13,14]. Thus, developing strategies to determine muscle protein synthesis and degradation are, therefore, necessary during muscle injury and inflammation.

Proinflammatory cytokine activity is one of the most important molecular mechanisms contributing to impaired muscle regeneration [15,16]. Chronic elevations in proinflammatory cytokines after severe injury contribute to impaired muscle homeostasis and myogenesis and lead to disuse atrophy [17,18]. As a mediator of inflammation, interleukin (IL)-18 plays an essential role in the pathogenesis of autoimmune diseases such as myositis, rheumatoid arthritis, and atherosclerosis [19,20]. IL-18 reportedly participates in tissue destruction by elevating the production of nitric oxide and reactive oxygen intermediates in macrophages and neutrophils [21]. High serum levels of IL-18 have been found in patients with age-related muscle wasting disease [22]. However, the role of IL-18 in muscle atrophy or the changes of protein synthesis and degradation are not fully understood.

High-mobility group box-1 (HMGB1) is a ubiquitous nuclear protein with extracellular inflammatory cytokine activity [23]. HMGB1 mediates activation of the innate immune response, including chemotaxis and proinflammatory cytokine release [24], and is involved in functional disturbances of muscle weakness in patients with inflammatory myopathy [25]. In skeletal muscle atrophy, recent research indicates that the activation of HMGB1 receptors leads to muscle dysfunction [26]. The interaction of HMGB1 with toll-like receptor 4 (TLR4) induces muscle weakness and fatigue in patients with myositis [27], while the binding of HMGB1 to the receptor for advanced glycation end-products (RAGE) decreases survival in muscle-wasting mouse models [28]. Interestingly, HMGB1-induced activation of IL-18 synthesis depends on the inflammatory activation of RAGE in macrophages [29]. However, it is unknown whether the activation of IL-18 by HMGB1 during inflammation induces the alterations of atrophy markers and myogenic markers in myoblasts. Therefore, we sought to investigate the role of HMGB1-induced increases of IL-18 expression in inflammatory skeletal muscle myoblasts and the effectiveness of using the HMGB1/IL-18 signaling axis as a therapeutic target in a mouse model of glycerol-induced muscle injury (GIMI).

## 2. Materials and Methods

### 2.1. Cell Cultures

C2C12 and G7 myoblasts were purchased from the American Type Culture Collection (ATCC, Manassas, VA, USA) and maintained in humidified air at 37 °C with 5% CO_2_ and cultured in Dulbecco’s modified Eagle’s medium (DMEM) supplemented with 10% fetal bovine serum (Thermo-Fisher Scientific, Inc., Waltham, MA, USA) containing antibiotics (100 U/mL penicillin, 100 µg/mL streptomycin).

### 2.2. Quantitative Real-Time Polymerase Chain Reaction (qRT-PCR)

Total RNA was extracted from murine myoblast cells using TRIzol™ reagent (MDBio, Taipei, Taiwan). qRT-PCR analysis was conducted according to our previous report [30]. RNA concentration was measured using a NanoVue Plus™ spectrophotometer (Biochrom Ltd., Cambridge, UK). 1 μg of total RNA was reverse-transcribed to complementary DNA (cDNA), which was then synthesized by the MMLV reverse transcription system (Invitrogen, Carlsbad, CA, USA) and mixed with Fast SYBR^®^ Green Mix. Gene expression was quantified by the StepOnePlus™ Real-Time PCR System (Applied Biosystems, Foster City, CA, USA). Glyceraldehyde 3-phosphate dehydrogenase (GAPDH) served as the internal control, and all primers used in the qRT-PCR assays are reported in Appendix A.

### 2.3. Western Blot Analysis

Tibialis anterior (TA) muscle tissue was homogenized in 50 mM Tris pH 7.4, 150 mM NaCl, and 1% Triton X-100, in the presence of a mixture of protease inhibitors (Roche Applied Science) [31,32]. 30 μg of total proteins from tissue homogenates and myoblast cell lysate were separated by SDS-PAGE electrophoresis and then transferred to Immobilon^®^ PVDF membranes according to the method described in our previous study [30,33]. Following blocking with 5% skim milk diluted in a mixture of Tris-buffered saline with 0.1 % Tween-20 (TBST) for 1 h. Primary and secondary antibodies were applied as in Appendix A. Membranes were washed and incubated with enhanced chemiluminescence substrate. Signals were visualized with the ImageQuant^TM^ LAS 4000 biomolecular imager (GE Healthcare, Little Chalfont, UK).

### 2.4. Enzyme-Linked Immunosorbent Assay (ELISA)

Myoblasts were cultured in 6-well plates. Cell culture medium was then exchanged with serum-free DMEM medium. Cells were treated with HMGB1 (0, 1, 3 or 10 ng/mL) then incubated at 37 °C for 24 h. To examine downstream signaling pathways responsive to HMGB1 treatment, cells were pretreated for 30 min with the inhibitors or transfected with the small interfering RNAs (siRNAs) to stimulate myoblasts, before undergoing 24 h of HMGB1 treatment (10 ng/mL). Thereafter, all conditioned mediums were collected and quantified for protein secretion with the IL-18 enzyme-linked immunosorbent assay kit (R&D systems, Minneapolis, MN, USA), according to the manufacturer’s protocols.

### 2.5. Immunofluorescence Staining

C2C12 myoblast cells were grown to 80% confluence in a growth medium (GM), then switched to a differentiation medium (DM; DMEM supplemented with 2% horse serum, 100 U/mL penicillin, and 100 µL/mL streptomycin) for 5 days to induce myoblast differentiation [34], which was changed every 2 days. Cells were pretreated with IL-18 monoclonal neutralizing antibody for 30 min before undergoing 24 h of HMGB1 treatment (10 ng/mL) on Day 4 post-differentiation. Myoblast or myotube cells were fixed in 3.7% paraformaldehyde at room temperature for 30 min, permeabilized with 0.1% Triton™ X-100 for 10 min, and blocked with 1% bovine serum albumin (BSA) for 1 h at room temperature and labeled with 2 µg/mL of myosin skeletal muscle rabbit polyclonal antibody or rabbit anti-desmin primary antibody overnight at 4 °C. Goat anti-rabbit IgG (H + L) cross-adsorbed secondary antibody Alexa-Fluor^®^ 488 conjugated (Thermo Fisher Scientific, Hemel Hempstead, UK) was used at a concentration of 1:1000 in phosphate-buffered saline (PBS) containing 0.2% BSA for 1 h at room temperature for detection of myosin heavy chain (MyHC) and desmin in the cytoplasm. The nuclei were then stained with 4,6-diamidino-2-phenylindole (DAPI) for 15 min. Immunofluorescence-stained cells were examined with a fluorescent microscope (Carl Zeiss, Oberkochen, Germany).

### 2.6. Hematoxylin and Eosin Staining

Cultured C2C12 myoblasts and differentiated myotubes were fixed with 4% paraformaldehyde at room temperature for 20 min. According to the manufacturer’s instruction, these cells were washed twice with PBS-1X and then stained using the hematoxylin and eosin (H & E) staining kit (Sigma-Aldrich, St. Louis, MO, USA). C2C12 myotube morphology was observed under a Leica microscope (Leica microsystems, Wetzlar, Germany). The percentage of fusion index was calculated as the number of nuclei in myotubes divided by the total number of nuclei present in the observed field [35]. ImageJ software version 1.8.0_345 (Wayne Rasband and contributors National Institutes of Health, Bethesda, MD, USA, 2022) was used to quantify the fusion index, myoblast area, and the percentage of myoblast cell numbers divided by myoblasts in the field.

### 2.7. HMGB1 Knockdown in C2C12 Myoblast Cell Lines

Recombinant lentiviruses were produced by co-transfecting 293T cells (Thermo Fisher Scientific, Cat. No. R70007) with 0.5 mg of a short hairpin RNA-expressing plasmid (TRCN0000365913) with 0.2 mg of the packaging plasmid pCMVΔR8.91 and the VSV-G envelope glycoprotein expression plasmid (pMD.G), according to a previously described method [36]. All plasmids were obtained from the Addgene or National RNAi Core Facility at the Academia Sinica in Taiwan (Appendix A). Forty-eight hours after transfection, lentiviral particles carrying HMGB1 shRNA (shHMGB1) were isolated from the supernatant of 293T cells, then filtered through a 0.45 µm syringe filter and stored at −80 °C until use. A plaque assay using serial dilution was performed using murine C2C12 myoblasts to determine viral titers (in plaque-forming units [pfu]) of Lenti-shHMGB1, following previous methodology [37], which determined the Lenti-shHMGB1 viral titers for treatment purposes to be ~7.1 × 10^6^ pfu in the mouse model of skeletal muscle atrophy.

For the generation of stable cell lines, C2C12 cells were seeded at a density of 1 × 10^5^ cells/well in 12-well plates for 24 h. Growth media were removed and infected with shHMGB1 particles. After 24 h, stably infected clones were selected with puromycin (2 mg/mL). Knockdown efficacy of the Hmgb1 gene was confirmed by qRT-PCR and Western blot analysis.

### 2.8. Transfection and Luciferase Reporter Assay

C2C12 cells were seeded into 6-well plates until 90% confluent. SiRNAs against RAGE, p85, Akt, mTOR and c-Jun, and also the siRNA control, were mixed with Dharma FECT 1 transfection reagent (Thermo, Waltham, MA, USA) for 15 min in Opti-MEM^®^ medium (Thermo, Waltham, MA, USA). The cells were then transfected with different plasmids for 24 h according to the manufacturer’s instructions. Transfection efficiency was analyzed by western blot.

The AP-1 (activation protein-1) luciferase plasmid was obtained as previously reported [30]. Luciferase activity was evaluated using a luciferase assay kit. All results were obtained from 5 independent experiments.

### 2.9. Chromatin Immunoprecipitation-Quantitative Polymerase Chain Reaction (ChIP-qPCR) Assay

C2C12 myoblast cells were pretreated for 30 min with the inhibitors to stimulate the myoblasts before they were subjected to 24 h of HMGB1 treatment (10 ng/mL). The ChIP assay was carried out as previously described [38]. DNA immunoprecipitated by anti-c-Jun antibody was purified then extracted with phenol-chloroform. The purified DNA pellet was quantified by qPCR and normalized with the input DNA, which was performed in triplicate with SYBR green mix using the StepOnePlus™ Real-Time PCR System (Applied Biosystems, Foster City, CA, USA). Forward 5′-AACCTTGGACCTTTACCCTTT-3′ and reverse 5′-GTGCCCACAGAGAGACTGAA-3′ primers were used to amplify the IL-18 promoter region containing the c-Jun binding site.

### 2.10. Glycerol-Induced Muscle Injury Mouse Model

Eight-week-old male C57BL/6J mice were purchased from the National Laboratory Animal Centre (certified numbers: CMUIACUC-2021-139) in Taipei, Taiwan. They were handled with compassion and with the intention of minimizing suffering. They were kept in separate cages at a constant temperature of 22 ± 2 °C under 12:12 h light:dark cycles. The GIMI experiment was carried out as previously described [39]. Right and left TA muscles were intramuscularly injected with sterilized glycerol (70 µL of 50%, *v/v*) on Day 0. Following immunization, the mice were randomly separated into three groups (control, GIMI, or GIMI with HMGB1 shRNA treatment) and given twice-daily injections of 70 µL of HMGB1 shRNA (*n* = 10). We then used the rotarod machine (Singa Technology Corporation, Taoyuan city, Taiwan) to examine endurance movement at 40 revolutions per min (rpm) over 30 min. The time was recorded when each mouse fell off the rotating cylinder, and the mean of 3 trials was used as the outcome measurement.

### 2.11. Micro-Computed Tomography Analysis

Prior to micro-CT imaging, samples were stained with phosphotungstic acid (PTA) for 1 month. Micro-CT scans of mouse tibias and femurs were performed using an ex vivo micro-CT scanner, Bruker Skyscan 1272 (Bruker micro-CT, Kontich, Belgium) at 8.5 μm voxel resolution. The scanning used 70 kVp X-ray voltage, 142 μA current and 10 watts output, with a 0.5-mm aluminum filter. Image reconstruction was performed using graphics processing unit (GPU)-based reconstruction software, and GPU-Nrecon (Bruker micro-CT, Kontich, Belgium). The software corrected for ring artifacts and beam hardening. Reconstructed cross-sections were re-orientated and the region of interest was selected. We analyzed tibial muscle areas using 2 mm images (236 slices). The volume of interest was 2.0–4.0 mm below the growth plate. We analyzed femoral muscles area using 2 mm images (236 slices). The volume of interest was 3.0–5.0 mm below the growth plate. Thresholding, region of interest (ROI) selection and bone morphometric analyses were performed using CTAn software (Ver 1.20.8, Bruker micro-CT, Kontich, Belgium).

### 2.12. Immunohistochemistry and Dystrophin Staining

Immunohistochemistry (IHC) staining was performed using an IHC Kit (Sigma-Aldrich, St. Louis, MO, USA), according to the manufacturer’s instructions. Paraffin-embedded sections were prepared, mounted on silane-coated slides, deparaffinized in xylene, rehydrated in a graded alcohol series, and rinsed in deionized water. After antigen retrieval, intrinsic peroxidase activity was blocked by incubation with 3% H_2_O_2_. Non-specific antibody-binding sites were blocked using 3% BSA in PBS. Sections were then incubated with appropriately diluted primary antibodies at 4 °C overnight. Secondary antibody (biotin-labeled goat anti-rabbit IgG) was applied for 1 h at room temperature. Staining was detected with 3,3ʹ-diaminobenzidine tetrahydrochloride (DAB) and observed under a Leica microscope (Leica microsystems, Wetzlar, Germany). For dystrophin staining, after incubation with primary antibody, the sections were incubated with goat anti-rabbit IgG (H + L) cross-adsorbed secondary antibody Alexa-Fluor^®^ 594 conjugated (Thermo Fisher Scientific, Hemel Hempstead, UK). The dystrophin immunofluorescence-stained sections were then examined with TissueFAXS^®^ Spectra systems (TissueGnostics, Vienna, Austria). ImageJ software was used to quantify dystrophin-positive staining and IHC staining through a cross-sectional area of muscle fibers and densitometry, respectively.

### 2.13. Reagent and Resources

See Appendix A.

### 2.14. Statistical Analysis

Data in all figures are presented as the mean  ±  standard deviation (SD). Statistical analyzes were performed with GraphPad Prism software version 8.2.1 (GraphPad Software, La Jolla, CA, USA). Student’s *t*-test was used to compare the means between two groups. Statistical comparisons of more than three groups were performed using one-way analysis of variance (ANOVA) with Bonferroni’s post hoc test, and two-way ANOVA or multi-way ANOVA were used with more than two factors. The statistical criterion for a significant difference was set at *p* < 0.05.

## 3. Results

### 3.1. HMGB1 Induces Skeletal Muscle Atrophy, and Attenuates Myogenic Marker Expression in Myoblasts

To determine the effects of HMGB1 expression on skeletal muscle, we initially investigated the effects of HMGB1 treatment on C2C12 and G7 cells by assessing muscle atrophy with mRNA levels of MuRF-2, Atrogin-1, and Myostatin as well as myogenic markers with mRNA levels IGF-1, Pax-7, and MyoD. HMGB1 upregulated levels of mRNA expression in genes involved in skeletal muscle atrophy (Figure 1A) and protein expression (Figure 1C). HMGB1 also suppressed mRNA and protein levels of myogenic markers, including IGF-1, Pax-7, and MyoD (Figure 1B,D). The knockdown efficacy of Hmgb1 gene expression in C2C12 myoblasts (C2C12/shHMGB1) was confirmed by immunoblotting and qRT-PCR results (Figure 1E,F). Next, we examined whether HMGB1 affects the expression of muscle-specific protein markers MyHC and desmin in myoblasts. Immunofluorescence staining of C2C12/Control and C2C12/shHMGB1 cells showed that HMGB1 decreased levels of MyHC and desmin protein expression (Figure 1G–I). HMGB1 also reduced the numbers and proliferation of C2C12/Control cells (Figure 1J,K). These data suggest that HMGB1 promotes skeletal muscle protein degradation and attenuates the myogenesis process in myoblasts.

### 3.2. HMGB1 Inhibits Myoblast Differentiation via an IL-18-Dependent Mechanism

Inflammation is an important pathogenic factor, contributing to tissue damage of various organ diseases, including skeletal muscle dysfunction [40,41]. Abnormal activation of HMGB1 plays key roles in the development of inflammation-related diseases [42]. To screen for proinflammatory cytokines regulated by HMGB1, we initially measured levels of IL-18, IL-6, IL-1β, and tumor necrosis factor-α (TNF-α) mRNA expression in C2C12 and G7 myoblast cell lines after 24 h of HMGB1 stimulation. IL-18 mRNA levels were high (Figure 2A). Immunoblotting and ELISA detected significant dose-dependent increases of IL-18 protein secretion after HMGB1 treatment (Figure 2B,C). IL-18 appears to be involved in murine myoblast inflammation.

To examine whether HMGB1-induced IL-18 inflammation affects myogenesis [43], we first quantified IL-18 mRNA levels in C2C12 and G7 cells in GM or DM after 24 h of HMGB1 stimulation compared with no such treatment of GM or DM. We found that HMGB1 induced increases of IL-18 mRNA levels of both murine myoblasts in the GM and DM (Figure 2D). We then investigated whether IL-18 is involved in muscle synthesis markers. Blocking HMGB1-induced IL-18 expression with neutralizing IL-18 monoclonal antibody (mAb) reduced skeletal muscle mRNA-related atrophy and promoted myogenic marker expression (Figure 2E,F). We then observed the suppression of IL-18 mRNA expression in C2C12/shHMGB1 cells compared with C2C12/Control cells (Figure 2G). Levels of MyHC and desmin protein markers were significantly increased in C2C12/shHMGB1-differentiated cells compared with C2C12/Control-differentiated cells (Figure 2H). Immunofluorescence and H&E staining detected MyHC and desmin expression (Figure 2H–L), which confirmed the involvement of IL-18 in HMGB1-induced myoblast differentiation. Neutralizing IL-18 significantly rescued myoblast differentiation in both the C2C12/Control and C2C12/shHMGB1 cells, as indicated by the positive staining of MyHC and desmin markers, as well as by H&E staining (Figure 2H,K). The quantified results of immunofluorescence (Figure 2I,J) and H & E (Figure 2L) staining are shown by the fusion index. These results indicate that HMGB1 promotes inflammatory cytokine expression and attenuates myoblast differentiation via an IL-18-dependent mechanism.

### 3.3. RAGE/p85/Akt Signaling Is Involved in HMGB1-Induced Increases of IL-18 Expression

A recent study has demonstrated that the PI3K and Akt signaling pathways are associated with HMGB1-induced proinflammatory cytokine expression in inflammatory diseases [44]. We detected time-dependent phosphorylation of PI3K on tyrosine 458 (p85) and of Akt phosphorylation on serine 473 after HMGB1 treatment (10 ng/mL) (Figure 3A). Our data showed that RAGE serves as an upstream molecule of p85/Akt signaling (Figure 3B,C) and we found that p85 and Akt siRNAs effectively inhibited p85 and Akt protein expression (Appendix A). Pretreatment of C2C12 myoblast cells with neutralizing RAGE antibody, the p85 inhibitor (LY294002), the Akt inhibitor, and their siRNAs, all suppressed HMGB1-induced increases of IL-18 mRNA and protein expression, as well as their secretion (Figure 3D–F). These results indicate that RAGE/p85/Akt signaling is involved in HMGB1-induced increases of IL-18 expression.

### 3.4. mTOR Signaling Is Required for the HMGB1-Promotes IL-18 Expression

Incubating myoblast cells for 2 h with HMGB1 time-dependently promoted mTOR-signaling phosphorylation (Figure 4A). We confirmed that mTOR signaling occurs downstream of the RAGE/p85/Akt signaling pathway (Figure 4B). Pretreating cells with rapamycin or mTOR siRNAs inhibited HMGB1-induced increases of *IL-18* mRNA and protein expression (Figure 4C,D), while transfecting cells with mTOR siRNA reduced mTOR protein expression (Appendix A). ELISA results showed a significant reduction in IL-18 secretion after mTOR signaling was inhibited (Figure 4E). These data suggest that HMGB1 activates mTOR through the RAGE/p85/Akt signaling pathway and thus increases IL-18 expression.

### 3.5. HMGB1-Induced Increases of IL-18 Expression in C2C12 Myoblasts Requires c-Jun Phosphorylation

Since AP-1 plays an essential role in inflammatory diseases [45], we examined whether AP-1 influences HMGB1-induced modulation of IL-18 expression in myoblast cells. Significant increases of c-Jun phosphorylation after HMGB1 treatment (Figure 5A) were reduced by the RAGE neutralizing antibody, LY294002, the Akt inhibitor, and rapamycin (Figure 5B). Transfecting C2C12 cells with c-Jun siRNA confirmed the effectiveness of the siRNA in Western blotting (Appendix A). Treating the cells with tanshinone IIA or transfecting them with c-Jun siRNA prior to HMGB1 treatment downregulated levels of IL-18 mRNA (Figure 5C) and IL-18 protein expression (Figure 5D), and protein secretion (Figure 5E). We then used the luciferase reporter to determine whether the RAGE, p85, Akt, mTOR, and c-Jun signaling cascades were involved in HMGB1-induced AP-1 transcription activity (Figure 5F,G). Significant increases of AP-1 promoter activity in C2C12 cells after HMGB1 treatment were prevented when the cells were pretreated with the RAGE neutralizing antibody, LY294002, the Akt inhibitor, rapamycin, or tanshinone IIA (Figure 5F), or co-transfected with AP-1 and the RAGE, p85, Akt, mTOR or c-Jun siRNAs (Figure 5G). We continued to check the c-Jun binding site in the IL-18 promoter region. The results of the ChIP-qPCR assay showed that HMGB1-induced binding of c-Jun to IL-18 promoter was reduced by the RAGE neutralizing antibody, LY294002, the Akt inhibitor, and rapamycin (Figure 5H). HMGB1 appears to promote IL-18 expression and requires c-Jun to activate RAGE/p85/Akt/mTOR signaling.

### 3.6. Inhibition of HMGB1 Rescues In Vivo Skeletal Muscle Injury

We then investigated the effects of HMGB1 shRNA treatment upon skeletal muscle in C57BL/6J mice after glycerol-induced muscle injury. TA muscles were injected with glycerol (70 µL of 50%, *v/v*) on Day 0 and other treatment of HMGB1 shRNA (70 µL) was injected into the TA muscle on Days 1, 3, 5, and 7. The mice were sacrificed and analyzed on Days 9 (Figure 6A). Body weights did not differ significantly among the study groups (Figure 6B). Rotarod performance was worse in the GIMI group than in the control group, whereas rotarod performance was significantly improved in the GIMI group treated with HMGB1 shRNA compared with the untreated GIMI group (Figure 6C). H&E staining revealed smaller hind limb muscle morphology and damaged muscle fibers in the GIMI mice compared with the control mice (Figure 6D,H). Micro-CT images of TA muscle after PTA staining showed that HMGB1 shRNA reversed GIMI reductions in TA muscle thicknesses and total volume of muscle and bone (Figure 6E,F). Western blot and IHC staining revealed significant increases of levels of HMGB1 and IL-18 protein expression in TA muscle and decreases of levels of MyHC and desmin protein in the GIMI group compared with the control group (Figure 6G–L). HMGB1 shRNA significantly inhibited the expression of IL-18 and increased MyHC and desmin expression in the GIMI group treated with HMGB1 shRNA (Figure 6G–L). Compared with untreated GIMI mice, immunofluorescent staining showed that HMGB1 shRNA significantly recovered dystrophin protein expression in muscle fibers (Figure 6M) and increased TA muscle fiber size (Figure 6N). It appears that inhibiting HMGB1 rescues skeletal muscle differentiation that is associated with IL-18 and improves muscle fiber morphology.

## 4. Discussion

The release of proinflammatory cytokines changes skeletal muscle homeostasis following muscle damage [46]. Prolonged exposure to proinflammatory cytokines following muscle injury may impair muscle protein homeostasis [47]. Several research groups have explored the associations between aberrant HMGB1 expression and muscle inflammation, weakness, and muscle fiber dysfunction [25,48,49]. Recent reports have shown that necroptosis in muscle cells contributes to the release of HMGB1 and thus accelerates muscle inflammation and subsequent muscle injury [50,51]. Activation of HMGB1/autophagy signaling is related to denervation-induced skeletal muscle atrophy [52], and the HMGB1/atrogin-1 axis has been shown to cause skeletal muscle dysfunction [53]. Interestingly, some evidence suggests that fully reduced HMGB1 induces muscle regeneration [54] and liver regeneration [55], whereas oxidized HMGB1 might limit muscle regeneration [56]. Thus, fully reduced and oxidized HMGB1 may fulfil different functions in the maintenance of muscle homeostasis, promoting either inflammation or regeneration. However, the recombinant HMGB1 that we used in this study was not analyzed, which could be a limitation of this study and should therefore be examined in future research.

A previous study showed that removal of MuRF-1 reduces the development of skeletal muscular atrophy by 36% [57]. Another study showed mRNA levels of MuRF-2 was higher than MuRF-1 in response to 3-day differentiation [58]. Wai et al. reported that MuRF, atrogin-1, and myostatin are important muscle catabolic signals in muscle wasting [8]. Thus, our results showed that HMGB1 increased the expression of skeletal muscle atrophy markers (atrogin-1, MuRF-2, and myostatin) via IL-18 expression. A previous study revealed that a transition in MyHC isoforms occurs during myogenesis and muscle cell differentiation [59]. Our results showed HMGB1-regulated MyHC expression in myoblasts and differentiated myotube might associate with myogenesis and muscle differentiation and require to be further investigated in the future. Our study evidence supports the contention that a balance of these markers is essential for maintaining myogenesis, suggesting that HMGB1 regulates muscle atrophy through those various myogenic markers. Targeting HMGB1 may prevent skeletal muscle degradation. Based on our results of Figure 1E,J, *Hmgb1* knockdown efficiency is approximately 50% which had no significant effect on cell viability, though HMGB1 is a damage associated molecular pattern which released during cell death.

One research group has reported finding that the RAGE ligand, S100B, induces the hallmarks of cancer cachexia associated with experimental colon adenocarcinoma or Lewis lung carcinoma [28]. Another recent paper has reported that colon cancer exosomes containing HMGB1 can induce muscle atrophy via the TLR4/nuclear factor-κB (NF-κB) pathway and that serum levels of HMGB1 expression are upregulated in patients with colon cancer cachexia [49]. Both of those studies focused on the role of RAGE of HMGB1 in cancer cachexia-induced muscle atrophy [28,49]. Our results show that HMGB1-induced mediation of IL-18 signaling impairs the expression of myogenesis markers and inhibits skeletal muscle myoblast differentiation. Notably, increasing levels of HMGB1 and IL-18 expression have been reported in patients with calcific aortic valve disease [60]. In this study, we have described harmful in vitro and in vivo effects of HMGB1-induced IL-18 inflammation via the RAGE/p85/Akt/mTOR/c-Jun signaling cascades.

TLR4 and RAGE are the primary HMGB1 receptors that mediate the release of proinflammatory cytokines [61]. Upregulated levels of HMGB1 and RAGE have been found in the skeletal muscle of patients with inclusion body myositis [62]. Evidence has shown that the HMGB1-RAGE interaction promotes myoblast death [63]. Other research has demonstrated that targeting RAGE prevents muscle wasting [28]. RAGE is also involved in HMGB1-induced promotion of NF-κB activation, so it is implicated in the production of proinflammatory cytokines [64]. In our experiment, we confirmed the regulatory effect of RAGE upon IL-18 expression by blocking RAGE cellular signaling. Our results suggest that RAGE is involved in HMGB1-induced promotion of IL-18 production and thus induces increases of skeletal muscle atrophy marker expression in vitro and in vivo (Appendix A). In addition, the HMGB1-TLR4 pathways play an essential role in causing muscle fatigue in patients with myopathy because TLR4 is activated in mouse and human skeletal muscle fibers after HMGB1 stimulation [27]. Inhibition of TLR4 and HMGB1 signaling significantly reduced the expression of TNF-*α* and IL-6 cytokines in a myopathy mouse model [65], suggesting that the HMGB1/TLR4 axis may involve in skeletal muscle atrophy. In our study, immunoblotting demonstrates the effectiveness of RAGE or TLR4 following HMGB1 stimulation in C2C12 cells (Appendix A). RAGE and TLR4 are both candidates for HMGB1 receptors via IL-18-dependent mechanisms in muscle differentiation markers, but more research is needed to clarify the association between HMGB1 and TLR4 in skeletal muscle atrophy.

Impaired muscle regeneration is associated with muscle injuries or myopathy [66,67]. Previous research has shown that glycerol injury induces early fibrosis in rat skeletal muscle [68] and disrupts neuromuscular junctions in skeletal muscle [69]. Our in vivo results revealed the rapid appearance of early-stage muscle injury after glycerol treatment. Our study found that glycerol induced significant changes in muscular morphology and decreased skeletal muscle fiber size, suggesting that HMGB1-promoted alterations in IL-18 expression have an essential role in the early stages of muscle injury. Our in vitro results show that the mechanism of HMGB1 promotes IL-18 production in skeletal myoblasts and differentiated myotubes, and that mRNA levels of muscle synthesis-related markers were suppressed in myoblasts after neutralizing IL-18 secretion. These results suggest that inhibiting HMGB1 rescues muscle differentiation via an IL-18-dependent mechanism. However, since we could not exclude the effects of other proinflammatory cytokines on muscle differentiation (Figure 2A), further studies are needed to explain this assumption. Our in vivo work also shows that targeting HMGB1/IL-18 signaling rescues muscle-specific differentiation marker expression in glycerol-induced muscle injury, which indicates that targeting IL-18 is worthwhile for the treatment of skeletal muscle atrophy. More in vivo studies are needed to clarify the underlying mechanism by which IL-18 directly induces skeletal muscle atrophy.

Monocytes are inflammatory cells that release HMGB1 in response to injury and infection [70,71]. IL-18 is produced by neutrophils, macrophages, dendritic cells, and epithelial cells [72,73,74,75]. Based on these references and our in vivo IHC staining of TA muscle, HMGB1 and IL-18 might originate from skeletal muscle cells or other infiltrating immune cells in the inflammatory microenvironment. Unequivocal evidence is needed from future investigations. A limitation of our study is that we did not collect serum samples from the GIMI mouse model, due to its rapid induction of muscle injury and subsequent regeneration of muscle, which occur by the time of sacrifice. We also investigate the expression of MyoD and myogenin (Appendix A). Using a more appropriate method or a longer-term animal model to distinguish disused-related muscle atrophy or muscle regeneration, respectively, needs to be carefully considered in future investigations.

In conclusion, we are the first to demonstrate that HMGB1-induced increases of IL-18 expression lead to skeletal muscle atrophy in vitro and in vivo (Figure 7). Our data suggest that the HMGB1/IL-18 axis could serve as an essential therapeutic target in muscle atrophy.

## Figures and Tables

**Figure 1 cells-11-03936-f001:**
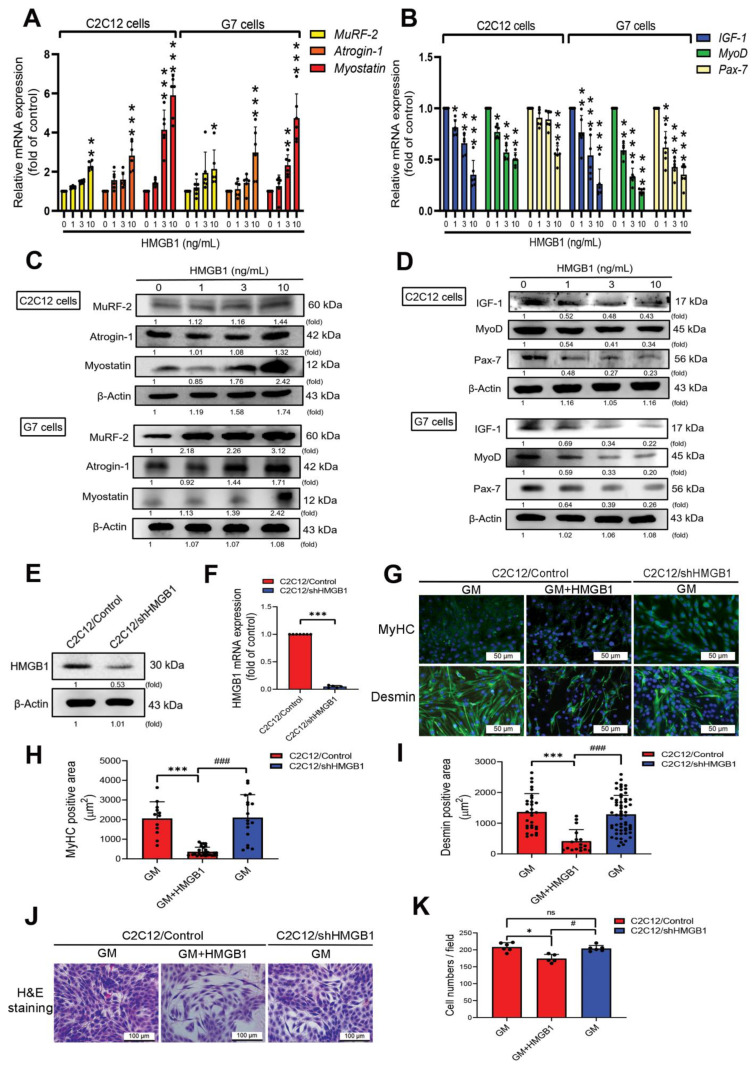
HMGB1 attenuates myogenic marker expression and muscle differentiation in myoblasts. (**A**–**D**) Levels of mRNA and protein expression of muscle atrophy and myogenic markers in C2C12 and G7 cell lines after 24 h of treatment with HMGB1 (0, 1, 3 or 10 ng/mL) (*n* = 6). β-Actin was used as a loading control. ImageJ software was used to quantify the immunoblotting results, which were normalized with β-actin. (**E**,**F**) Data show the knockdown efficacy of HMGB1-shRNA detected by (**E**) immunoblotting and (**F**) qRT-PCR (*n* = 6). (**G**) Immunofluorescence staining detected skeletal muscle markers MyHC and desmin, represented by the green color. Cell nuclei were detected by DAPI staining, represented by the blue color; scale bar = 50 µm. Quantified area results of (**H**) MyHC (*n* = 12) and (**I**) desmin-positive staining in C2C12 (*n* = 20) were calculated by ImageJ software. (**J**,**K**) Hematoxylin and eosin (H & E) staining results were quantified by the ratios between myoblast cell number divided by the field in growth medium (GM) using ImageJ software. Scale bar = 100 µm. Results are expressed as the means  ±  SD of three independent experiments. Statistical analysis was done by using student’s *t*-test was used to compare the means between two groups, one-way ANOVA was used to compare the means among three or more groups, and two-way ANOVA or multi-way ANOVA were used with more than two factors. * *p* < 0.05; ** *p* < 0.01; *** *p* < 0.001 compared with the control group, or the group that did not receive HMGB1 treatment; # *p* < 0.05; ### *p* < 0.001 compared with the HMGB1-treated group.

**Figure 2 cells-11-03936-f002:**
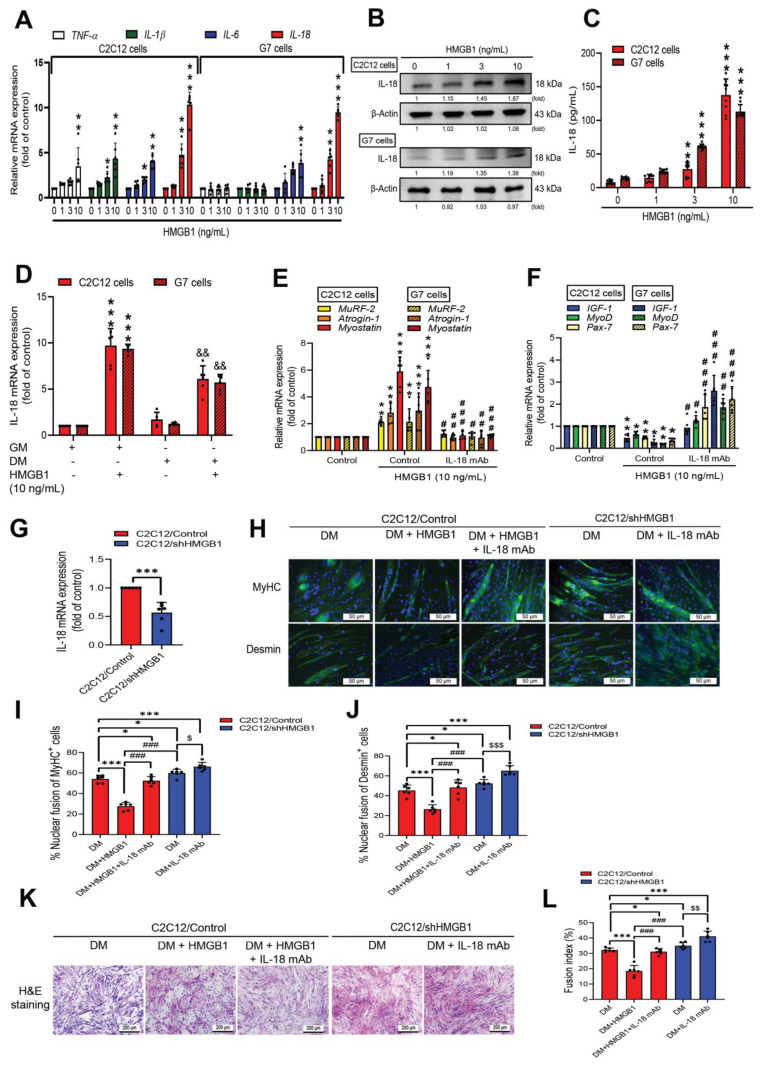
HMGB1 inhibits myoblast differentiation via an IL-18-dependent mechanism. (**A**) Level of mRNA expression of inflammatory cytokines in C2C12 and G7 myoblast cells after treatment with HMGB1 (0, 1, 3 or 10 ng/mL). Results are expressed as the means  ±  SD of five independent experiments (*n* = 6). (**B**) C2C12 and G7 cells were treated with HMGB1 (0, 1, 3 or 10 ng/mL) for 24 h, before immunoblotting the cell lysates to detect intracellular IL-18 expression. β-Actin was used as a loading control. ImageJ software was used to quantify the immunoblotting results, which were normalized with β-actin. (**C**) ELISA was used to detect levels of secreted IL-18 expression in culture medium (*n* = 8). (**D**) Levels of IL-18 mRNA expression in C2C12 or G7 cells cultured in GM or differentiation medium (DM) with or without HMGB1 (10 ng/mL) stimulation (*n* = 6). Levels of mRNA expression in (**E**) muscle atrophy and (**F**) myogenic markers in C2C12 or G7 myoblast cells after treatment with HMGB1 (10 ng/mL) or anti-IL-18 monoclonal antibody (30 ng/mL) for 24 h (*n* = 6). (**G**) Levels of *IL-18* mRNA expression in C2C12/shHMGB1 cells were compared with levels in C2C12/Control cells (*n* = 6). (**H**) Immunofluorescence staining detected the expression of skeletal muscle markers MyHC and desmin in the C2C12/Control and C2C12/shHMGB1 groups. MyHC and desmin are represented by the color green. Cell nuclei was detected by DAPI staining in the color blue; scale bar = 50 µm. (**K**) Morphology of myotube formation was detected by hematoxylin and eosin (H&E) staining; scale bar = 200 µm. Quantified analysis by ImageJ software shows the nuclear fusion of (**I**) MyHC- and (**J**) desmin-positive staining or (**L**) the fusion index in the differentiation medium. Error bars indicate means ± SD. Statistical analysis was done by using student’s *t*-test was used to compare the means between two groups, one-way ANOVA was used to compare the means among three or more groups, and two-way ANOVA or multi-way ANOVA were used with more than two factors. * *p* < 0.05; ** *p* < 0.01; *** *p* < 0.001 compared with the control group or the C2C12/Control group; **#**
*p* < 0.05; ## *p* < 0.01; ### *p* < 0.001 compared with the HMGB1-treated group; && *p* < 0.05 compared with the C2C12 or G7 cell line in DM that did not receive HMGB1 treatment; $ *p* < 0.05; $$ *p* < 0.01; $$$ *p* < 0.001 compared with the C2C12/shHMGB1 control group.

**Figure 3 cells-11-03936-f003:**
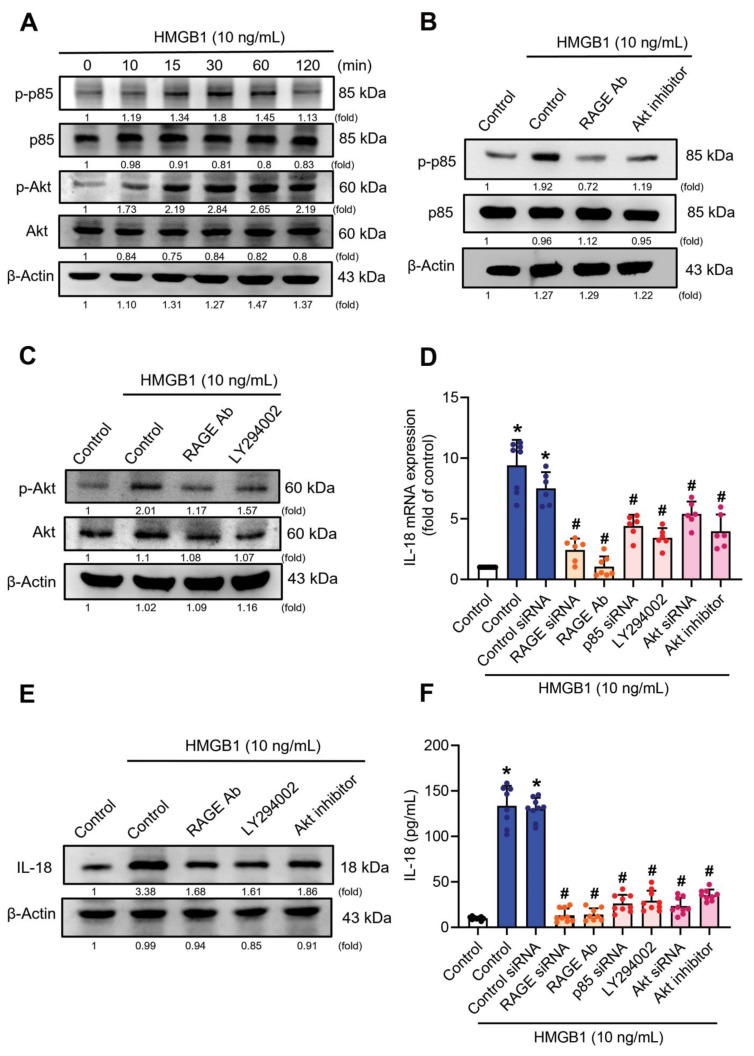
The RAGE/p85/Akt signaling pathway is involved in HMGB1-induced upregulation of IL-18 expression. (**A**) C2C12 cells were treated with 10 ng/mL of HMGB1. Cell lysate was collected at the indicated time points and subjected to immunoblotting to detect total protein levels and phosphorylation status of the indicated proteins. (**B**,**C**) C2C12 cells were pretreated with anti-RAGE neutralizing antibody (5 µg/mL), Akt inhibitor (10 µM) or LY294002 (10 µM) for 30 min, then treated with HMGB1 (10 ng/mL) for 30 min before quantifying protein expression and phosphorylation levels of the p85 or Akt proteins by immunoblotting. (**D**–**F**) C2C12 cells were transfected with either control siRNA, RAGE siRNA, p85 siRNA, or Akt siRNA for 24 h and then treated with HMGB1 (10 ng/mL) for 24 h, or C2C12 cells were pretreated with RAGE neutralizing antibody (5 µg/mL), LY294002 (10 µM) or Akt inhibitor (10 µM) for 30 min followed by HMGB1 (10 ng/mL) treatment for 24 h. Levels of IL-18 mRNA and protein expression, as well as IL-18 secretion, were quantified by (**D**) qRT-PCR (*n* = 8), (**E**) immunoblot, and (**F**) ELISA (*n* = 8), respectively. Results are expressed as the means  ±  SD of three independent experiments. One-way ANOVA was used to compare the means among three or more groups. * *p* < 0.05 compared with controls group; # *p* < 0.05 compared with the HMGB1-treated group alone. ImageJ software was used to quantify the immunoblotting results and normalized with β-actin.

**Figure 4 cells-11-03936-f004:**
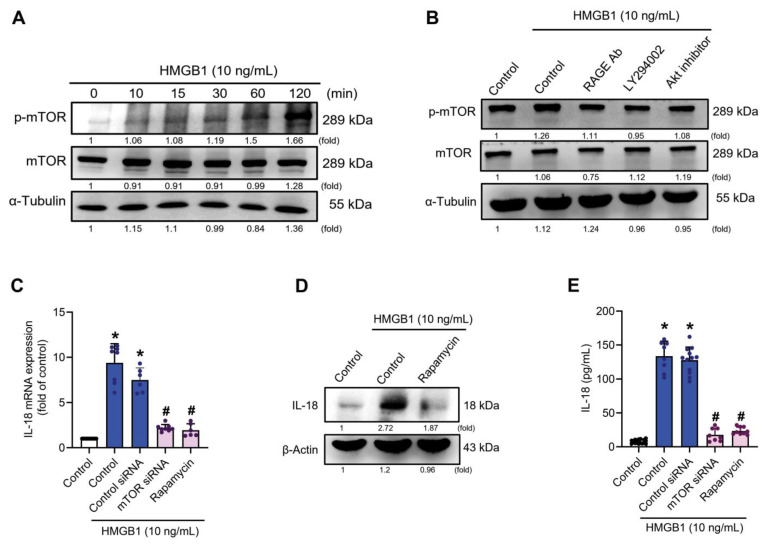
The mTOR signaling pathway is involved in HMGB1-induced upregulation of IL-18 expression. (**A**) C2C12 cells were treated with 10 ng/mL HMGB1, then cell lysate was collected at the indicated time points and subjected to immunoblotting to detect total protein levels and phosphorylation levels of indicated proteins. α-Tubulin was used as a loading control. (**B**) C2C12 cells were pre-treated with RAGE neutralizing antibody (5 µg/mL), LY294002 (10 µM) or Akt inhibitor (10 µM) for 30 min, then with HMGB1 (10 ng/mL) for 2 h. Immunoblotting detected protein expression and phosphorylation levels of the indicated proteins. (**C**–**E**) IL-18 mRNA and protein expression, and secretion levels were detected by (**C**) qRT-PCR (*n* = 6), (**D**) immunoblotting and (**E**) IL-18 ELISA (*n* = 8), respectively. Results are expressed as the means  ±  SD of three independent experiments. One-way ANOVA was used to compare the means among three or more groups. * *p* < 0.05 compared with the control group; # *p* < 0.05 compared with the HMGB1-treated group alone. ImageJ software was used to quantify the immunoblotting results, which were normalized with α-tubulin or β-actin.

**Figure 5 cells-11-03936-f005:**
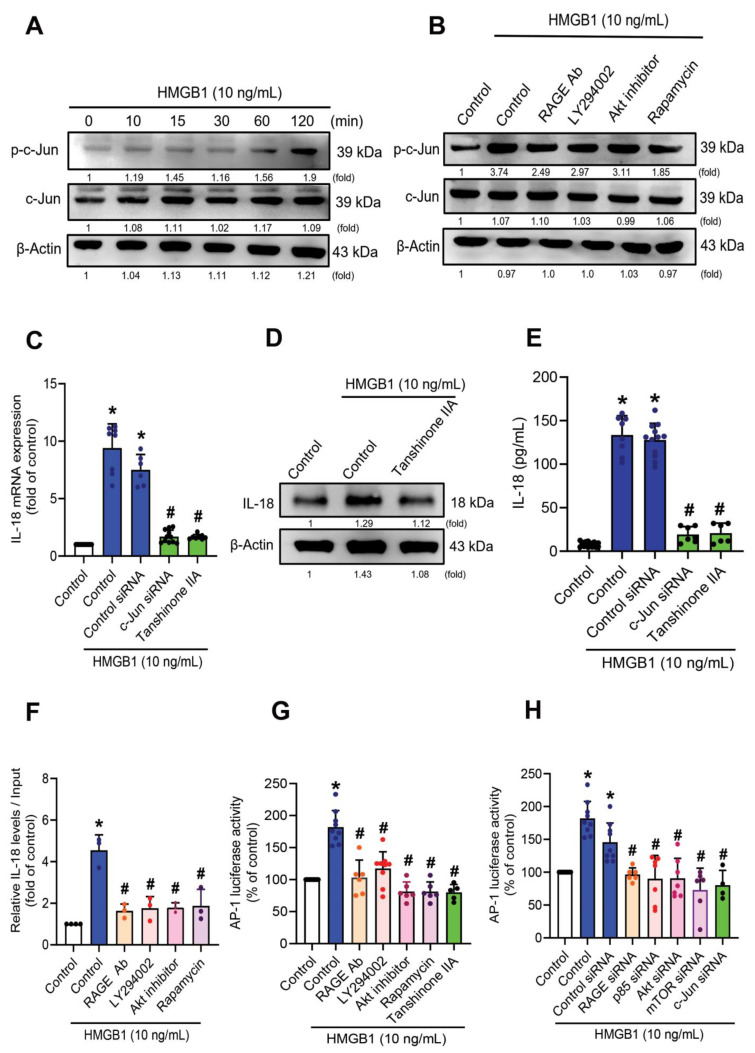
c-Jun is involved in HMGB1-induced IL-18 production via the RAGE/p85/Akt/mTOR signaling pathway. (**A**) C2C12 cells were treated with HMGB1 (10 ng/mL). Cell lysate was collected at the indicated time points and subjected to immunoblotting to detect total protein levels and phosphorylation levels of the indicated proteins. (**B**) C2C12 cells were pretreated with RAGE neutralizing antibody (5 µg/mL), LY294002 (10 µM), Akt inhibitor (10 µM), or rapamycin for 30 min, then with HMGB1 (10 ng/mL) for the next 2 h, before detecting protein expression and phosphorylation levels of the indicated proteins. (**C**–**E**) The C2C12 cells were transfected with control siRNA or c-Jun siRNA, or pretreated with tanshinone IIA (10 µM), followed HMGB1 (10 ng/mL) treatment for 2 h. IL-18 mRNA expression, protein expression and levels of IL-18 secretion were detected by (**C**) qRT-PCR (*n* = 8), (**D**) immunoblotting and (**E**) IL-18 ELISA (*n* = 10), respectively. (**F**) IL-18 activation was examined by chromatin immunoprecipitation (*n* = 3). (**G**–**H**) The activity of the AP-1-luciferase plasmid was increased, and the results were normalized to β-galactosidase activity. (**G**) Cells were treated with the RAGE neutralizing monoclonal antibody, LY294002, Akt inhibitor, rapamycin, or tanshinone IIA for 30 min, then treated with HMGB1 for 24 h (*n* = 6). (**H**) Cells were co-transfected with the AP-1-luciferase plasmid and siRNAs against RAGE, p85, Akt, mTOR or c-Jun for 24 h, then stimulated with HMGB1 (*n* = 6). All results are expressed as the means  ±  SD of three independent experiments. *****
*p* < 0.05 compared with the control group; # *p* < 0.05 compared with the HMGB1-treated group alone. ImageJ software was used to quantify the immunoblotting results, which were normalized with β-actin.

**Figure 6 cells-11-03936-f006:**
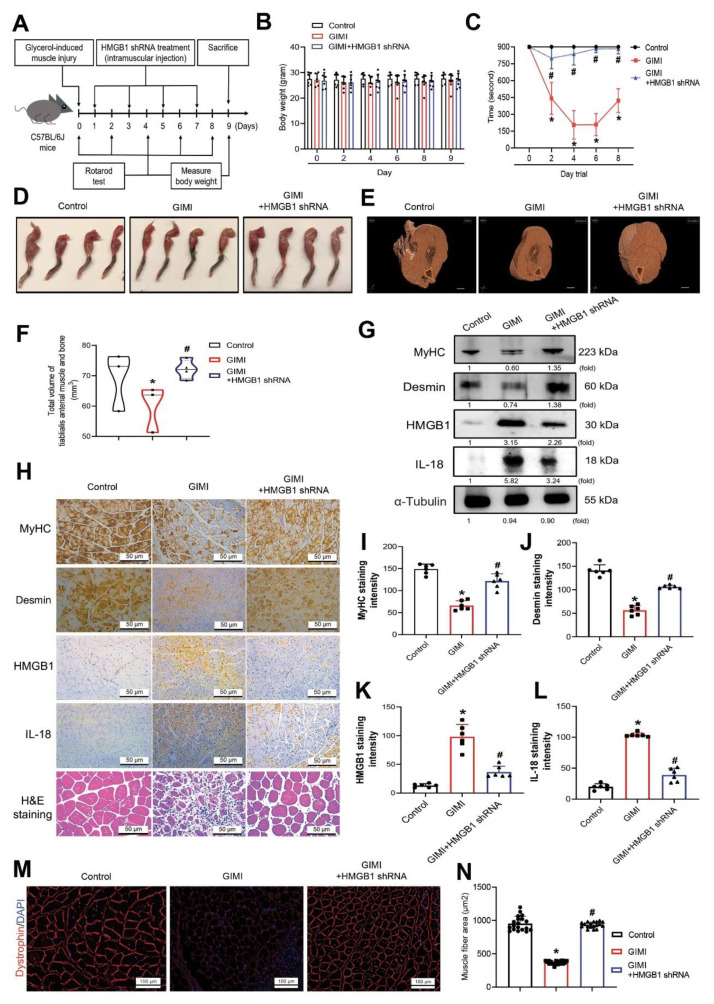
Downregulating HMGB1 rescues skeletal muscle atrophy in the glycerol-induced muscle injury (GIMI) mouse model. (**A**) The mice were randomly separated into three groups: control, GIMI, or GIMI + HMGB1 shRNA (*n* = 10 mice/group). Muscle injury was induced in 8-week-old male C57BL/6J mice via intramuscular injection of glycerol (70 µL of 50%, *v/v*) into tibialis anterior (TA) muscle on Day 0. Mice received a single injection of HMGB1 shRNA (70 µL) into the TA muscle on Days 1, 3, 5, and 7. (**B**) Bodyweight was measured on Days 0, 2, 4, 6, 8, and 9. (**C**) Muscle function was evaluated by rotarod testing (*n* = 10). (**D**) The gross hindlimb musculature of mice in the control group, GIMI group and GIMI with HMGB1 shRNA treatment group. (**E**) Micro-CT scan of TA muscle after PTA staining. (**F**) Total volume of TA muscle and bone were evaluated by micro-CT (*n* = 3). (**G**) Western blot identification of MyHC, desmin, HMGB1, and IL-18 protein from TA muscle tissue lysates of the control group, GIMI group and GIMI + HMGB1 shRNA treatment group. α-Tubulin was used as a loading control. (**H**) Immunohistochemistry analysis was performed to detect the expression of MyHC, desmin, HMGB1, and IL-18 (magnification 40×) and representative TA muscle cross-sections stained with H&E. Scale bar = 50 µm. (**I–L**) The immunohistochemical staining of (**I**) MyHC, (**J**) desmin, (**K**) HMGB1 and (**L**) IL-18 protein were quantified using densitometric analysis by ImageJ software (*n* = 6). (**M**) Immunostaining shows restoration of dystrophin in the TA muscle. Dystrophin is shown in red color. Nuclei are marked by DAPI staining in blue color. Scale bar = 100 µm. (**N**) Quantification of cross-section area of the TA muscles after dystrophin staining (*n* = 10). Results are expressed as means ± SD. One-way ANOVA was used to compare the means among three or more groups. * *p* < 0.05 compared with the control group, # *p* < 0.05 compared with the GIMI group.

**Figure 7 cells-11-03936-f007:**
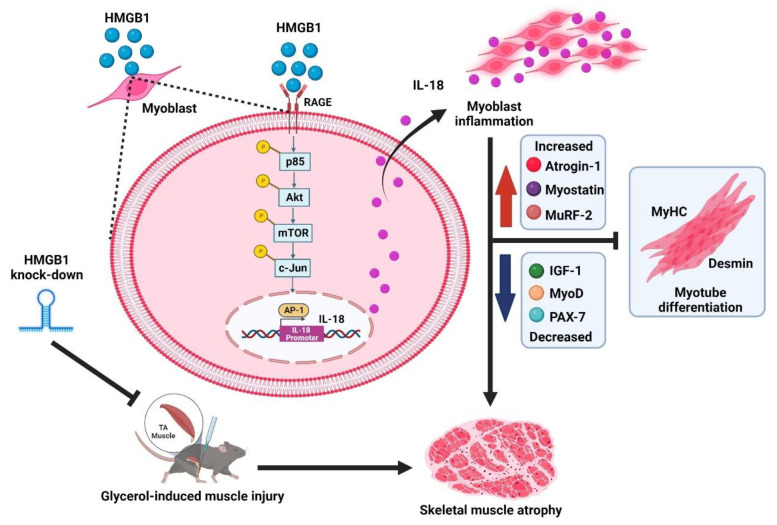
Schematic diagram illustrating the in vitro and in vivo effects of the HMGB1/IL-18 axis upon skeletal muscle atrophy.

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
