# Peer review of "HMGB1 Promotes In Vitro and In Vivo Skeletal Muscle Atrophy through an IL-18-Dependent Mechanism"

_cells, 2022, doi:10.3390/cells11233936_

Round 1

Reviewer 1 Report

The study by Ho and colleagues tests the hypothesis that HMGB1 is involved in skeletal muscle atrophy, and implicates IL-18 as a potential mechanism. Overall I was interested to read the paper and the findings were insightful, but I felt that the results were somewhat overstated and the experiments should be more clearly explained (and potentialy interpretted).

The introduction lacks clarity in the writing, and introduces undefined terms. I have some doubt as to whether the authors understand the differences between atrophy and impaired regeneration, as these are not the same things and the introduction fails to adequately distinguish between them. The statements about pro-inflammatory cytokines role in skeeltal muscle atrophy is quite sweeping because there are certainly some examples (e.g. disuse) where cytokine storms are not the likely culprit in driving muscle wasting (i.e. decreased synthesis rather than breakdown). Overall the introduction fails to paint a clear picture and set the scene for the work to be conducted.

there are some small issues with the methods that should also be tended to:

- how much protein was loaded into western blots?

- where was horse serum purchased from?

-line 168.. should this be siRNAs against RAGE?

- which plasmids were used?

- the statistical analysis section states that all quantified results were analysed using a students t-test... but this is not appropraite for all comparisisons? for examplefiguire 1 H, I and K should be analsysed using a one-way anova (if normally distributed).

Figure 1 reports markers of muscle breakdown such as Myostatin and Atrogin-1. Why are these experiments conducted in myoblasts when it seems more relvant to do this work in myotubes. You also measure MyHC, but this protein is really only expressed highly in myotubes- so that would have been a much better model to use. Similarly, the markers for muscle regeneration are cherry-picked, but there is not much rationale provided for this (a line or two in the intro). why not other MRFs such as myogenin and Myf-5? how were the images quantified; this seems difficult/impossible by eye.

All figures legens require a note about the sample size (n=?). Some of the inhibitor experiments (e.g. Fig 3b) do not show a huge reduction in protein phosphorylation using inhibitors and it is important to know how consistent this is. if you just did it once then 1.16 vs 0.83 might not really mean anything. 

line 365- AP1 needs defining

The text description of figure 6 needs improving. The authors should consider altering GIMA for GIMI, since the atrophy is induced by injury. the text should include some statemtns about the timings of this experiment. how long after you injected glycerol did you take these measurements? I am surprised that there are no good markers of msucle regernation included here, such as tracking satellite cells or measuring mRNA levels of myogenic markers such as MyoD or Myogenin. These markers would make more convincing argument about regeneration. I also think it is an overstatement to say that "inhibiting HMGB1 induced increases in IL-18 rescue...differentiation", becuse all you have there is an assocaition. As you have shown in the previous experiments, HMGB1 regulates signalling pathways such as Akt-mTOR which do more than just drive IL-18 expression. It is safer to say that inhibiting HMGB1 rescues differentiation and that it is assocaited with IL-18. 

Author Response

Point-by-point reply to the reviewers:

Reviewer 1:

Q1. The introduction lacks clarity in the writing, and introduces undefined terms. I have some doubt as to whether the authors understand the differences between atrophy and impaired regeneration, as these are not the same things and the introduction fails to adequately distinguish between them.   Overall the introduction fails to paint a clear picture and set the scene for the work to be conducted.

Authors’ response: We appreciate the constructive comments from this reviewer. We are very sorry for making reviewer misunderstanding the unclear Introduction.  Therefore, we delete the redundant information regarding regeneration that we intend to focus on muscle injury-related atrophy, and we also totally modify and rewrite the Introduction section:

“Muscle atrophy derives from musculoskeletal disuse following muscle injury [6] Muscle injury-related disuse is associated with the changes in muscle protein synthesis and degradation [7]. Molecular investigations into protein degradation related to skeletal muscle loss have focused on the ubiquitin ligase atrogin-1 [8], on muscle RING-finger-1 (MuRF-1) and MuRF-2 as markers of skeletal muscle atrophy [9], and on myostatin as a biomarker of muscle wasting [10]. The myogenic markers myogenic differentiation (MyoD), paired box 7 (Pax-7), and insulin-like growth factor 1 (IGF-1) promote muscle protein synthesis and maintain muscular function [11, 12]. In addition, an abundance of inflammatory cytokines in response to muscle injury may promote alterations of muscle atrophy markers and myogenic markers [13, 14]. Thus, developing strategies to determine muscle protein synthesis and degradation are therefore necessary during muscle injury and inflammation.” (Line 44─55)

“Proinflammatory cytokine activity is one of the most important molecular mechanisms contributing to impaired muscle regeneration [15, 16]. Chronic elevations in proinflammatory cytokines after severe injury contribute to impaired muscle homeo-stasis and myogenesis and lead to disuse atrophy [17, 18].” (Line 56─59)

Q2. How much protein was loaded into western blots?

Authors’ response: We appreciate the suggestion from this reviewer. In this study, we loaded 30 μg of total protein into the western blotting experiments. We have accordingly added the following information to the Materials and methods sections:

“30 μg of total proteins from tissue homogenates and myoblast cell lysate were…” (Line 104)

Q3. Where was horse serum purchased from?

Authors’ response: We appreciate the suggestion from this reviewer. We purchased the horse serum from Gibco company, and the identifier of this horse serum product is Cat#16050-122. We have accordingly added the following information to the Supplementary Tables S2.

Q4. Line 168.. should this be siRNAs against RAGE?

Authors’ response: We appreciate the suggestion from this reviewer. We have accordingly modified in the Materials and methods section:

“SiRNAs against RAGE, p85, Akt, mTOR and c-Jun…” (Line 167)

Q5.- Which plasmids were used?

Authors’ response: We appreciate the suggestion from this reviewer. In this study, we used mouse cytomegalovirus (CMV) promoter plasmid (Cat#169739), and myeloid differentiation factor 2 (MD-2) plasmid (Cat#20864) were purchased from Addgene company. And mouse shRNA HMGB1 plasmid (Cat#TRCN0000365913) was purchased from National RNAi Core Facility (Sinica, Taiwan). We have accordingly added the following information to the Materials and methods and Supplementary Tables S2.

“All plasmids were obtained from the Addgene or National RNAi Core Facility at the Academia Sinica in Taiwan (Supplementary Tables S2).” (Line 153─154)

Q6. The statistical analysis section states that all quantified results were analysed using a students t-test... but this is not appropraite for all comparisisons? for example figuire 1 H, I and K should be analsysed using a one-way anova (if normally distributed).

Authors’ response: We appreciate the comments from this reviewer. We have accordingly used one-way ANOVA to re-calculate again and modified all legends. We have accordingly added the following information to the Materials and methods sections:

“Student’s t-test was used to compare the means between two groups. Statistical com-parisons of more than three groups were performed using one-way analysis of variance (ANOVA) with Bonferroni’s post hoc test, and two-way ANOVA or multi-way ANOVA were used with more than two factors.” (Line 235─238)

Q7. Figure 1 reports markers of muscle breakdown such as Myostatin and Atrogin-1. Why are these experiments conducted in myoblasts when it seems more relvant to do this work in myotubes. You also measure MyHC, but this protein is really only expressed highly in myotubes- so that would have been a much better model to use. Similarly, the markers for muscle regeneration are cherry-picked, but there is not much rationale provided for this (a line or two in the intro). why not other MRFs such as myogenin and Myf-5? how were the images quantified; this seems difficult/impossible by eye.

Authors’ response: We appreciate the constructive comments from this reviewer. Previous study has shown the results of Myostatin and Astrogin-1 in myocytes (J. Cachexia Sarcopenia Muscle, 2020; 11: 120–134). We have shown the expression of MyHC in both myoblast (Figure 1G) and differentiated myotubes (Figure 2H). We also rewrite the introduction according to the first comment (Q1) above. We also investigate the expression of MyoD and myogenin (Supplementary Figures S4). We also change the blurred image in Figure 1 and re-calculate the quantification. We accordingly included the detailed information in the Discussion section:

“We also investigate the expression of MyoD and myogenin (Supplementary Figures S4). Using a more appropriate method or a longer-term animal model to distinguish dis-used-related muscle atrophy or muscle regeneration, respectively, needs to be carefully considered in future investigations.” (Line 525─528)

Q8. All figures legens require a note about the sample size (n=?). Some of the inhibitor experiments (e.g. Fig 3b) do not show a huge reduction in protein phosphorylation using inhibitors and it is important to know how consistent this is. if you just did it once then 1.16 vs 0.83 might not really mean anything.

Authors’ response: We appreciate the comments from this reviewer. We have accordingly modified all figure legends and changed Figure 1C, 3B,6G.

Q9. line 365- AP1 needs defining

Authors’ response: We appreciate the comments from this reviewer. Activator protein 1 (AP-1) has been initially shown at Methods in the original manuscript (Line 174).

Q10. The text description of figure 6 needs improving. The authors should consider altering GIMA for GIMI, since the atrophy is induced by injury. the text should include some statemtns about the timings of this experiment. how long after you injected glycerol did you take these measurements? I am surprised that there are no good markers of msucle regernation included here, such as tracking satellite cells or measuring mRNA levels of myogenic markers such as MyoD or Myogenin. These markers would make more convincing argument about regeneration. I also think it is an overstatement to say that "inhibiting HMGB1 induced increases in IL-18 rescue...differentiation", becuse all you have there is an assocaition. As you have shown in the previous experiments, HMGB1 regulates signalling pathways such as Akt-mTOR which do more than just drive IL-18 expression. It is safer to say that inhibiting HMGB1 rescues differentiation and that it is assocaited with IL-18.

Authors’ response: We appreciate the constructive comments from this reviewer. We have accordingly changed GIMA into GIMI in the whole text and legends. We also measured the in vivo expression of MyoD and Myogenin (MyoG) in Supplementary Figures S4.  We accordingly included the detailed information in the Results and Discussion sections:

“TA muscles were injected with glycerol (70 µl of 50%, v/v) on Day 0 and other treatment of HMGB1 shRNA (70 µl) was injected into the TA muscle on Days 1, 3, 5, and 7. The mice were sacrificed and analyzed on Days 9 (Figure 6A).” (Line 401─404)

“It appears that inhibiting HMGB1 rescues skeletal muscle differentiation that is associated with IL-18 and improves muscle fiber morphology.” (Line 420─421)

“We also investigate the expression of MyoD and myogenin (Supplementary Figures S4). Using a more appropriate method or a longer-term animal model to distinguish dis-used-related muscle atrophy or muscle regeneration, respectively, needs to be carefully considered in future investigations.” (Line 525─528)

Reviewer 2 Report

This paper studied the mechanism of HMHB1 on the expression of IL-18 and its effect on the phenotype of skeletal muscle atrophy. The authors concluded that HMHB1 promotes skeletal muscle atrophy through IL-18. In this paper, the authors used in vivo and in vitro models, and a variety of signal pathway inhibitors. In general, this paper has some novelty. However, there are still several problems.

 Major issue

 (1) The main problem is the lack of direct evidence in vivo and in vitro of IL-18 induced muscle atrophy, which leads to the inconclusive conclusion that HMHB1 plays a role in promoting muscle atrophy through IL-18. In in vitro experiments, there is no evidence of IL-18 inducing cells atrophy. According to the results of in vivo experiments, it can not be determined that the effect on skeletal muscle atrophy is through the IL-18 pathway. Therefore, the conclusion of the topic cannot be reached.

 (2) Figure 6H stainings have problems. MyHc and Desmin staining should be in the muscle fiber, not on the muscle fiber membrane. The staining results of HMGB1 and IL8 also had similar problems. Please explain.

 (3) Statistical method description error. One way ANOVA should be used for 3 groups or more, and two way ANOVA or multi way ANOVA  should be used with more than two factors.

 (4) The result of AP-1 is quite abrupt. In this paper, the data of effect of HMHB1 on the expression of AP-1 was not shown, and the authors directly detected AP-1 luciferase activity. In theory, the authors should detect the impact of AP-1 on IL-18 promoter, and it would be more reasonable to detect IL-18 promoter luciferase activity.

 Minor

 (5) In Figure 6G, the result of tubulin need to be replaced with clearer blot.

(6) This paper used IL-18mAb for IL-18 signal inhibition. Because it is difficult for antibody molecules to enter cells to play a role. The author needs to explain the relationship between HMGB1 and IL-18, and whether IL-18 plays a role in cells or outside cells.

(7) In the animal model induced by Glycerol, the literature cited by the author was a model by Acrolein instead of Glycerol. Please explain why or cite a more appropriate literature. In addition, Glycerol mainly result in skeletal muscle injury. It is suggested that the author adopt a more appropriate animal model of skeletal muscle atrophy in the future.

(8) Please add a schematic diagram of possible signal mechanisms.

Author Response

Point-by-point reply to the reviewers:

Reviewer 2:

Q1. The main problem is the lack of direct evidence in vivo and in vitro of IL-18 induced muscle atrophy, which leads to the inconclusive conclusion that HMHB1 plays a role in promoting muscle atrophy through IL-18. In in vitro experiments, there is no evidence of IL-18 inducing cells atrophy. According to the results of in vivo experiments, it can not be determined that the effect on skeletal muscle atrophy is through the IL-18 pathway. Therefore, the conclusion of the topic cannot be reached.

Authors’ response: We appreciate the constructive comments from this reviewer. Our in vitro data showed a strong and significant result that we used IL-18 neutralizing antibody to confirm HMGB1-induced IL-18 expression. Also, we showed the evidence that “Blocking HMGB1-induced IL18 expression with neutralizing IL-18 monoclonal antibody (mAb) reduced skeletal muscle mRNA-related atrophy and promoted myogenic marker expression (Figure 2E-F).” (Line 304─306).

Our in vivo results showed that “inhibiting HMGB1 rescues skeletal muscle differentiation that is associated with IL-18 and improves muscle fiber morphology.” (Line 419─420).

Therefore, our conclusion demonstrated that HMGB1-induced increases in IL-18 expression lead to skeletal muscle atrophy in vitro and in vivo.

Q2. Figure 6H stainings have problems. MyHc and Desmin staining should be in the muscle fiber, not on the muscle fiber membrane. The staining results of HMGB1 and IL8 also had similar problems. Please explain.

Authors’ response: We appreciate the comments from this reviewer. We have accordingly re-stained all Figure 6H for providing better and consistent quantified results.

Q3. Statistical method description error. One way ANOVA should be used for 3 groups or more, and two way ANOVA or multi way ANOVA should be used with more than two factors.

Authors’ response: We appreciate the comments from this reviewer. We have accordingly used one-way ANOVA to re-calculate again and modified all legends. We have accordingly added the following information to the Materials and methods sections:

“Student’s t-test was used to compare the means between two groups. Statistical com-parisons of more than three groups were performed using one-way analysis of vari-ance (ANOVA) with Bonferroni’s post hoc test, and two-way ANOVA or multi-way ANOVA were used with more than two factors.” (Line 235─238)

Q4. The result of AP-1 is quite abrupt. In this paper, the data of effect of HMHB1 on the expression of AP-1 was not shown, and the authors directly detected AP-1 luciferase activity. In theory, the authors should detect the impact of AP-1 on IL-18 promoter, and it would be more reasonable to detect IL-18 promoter luciferase activity.

Authors’ response: We appreciate the constructive comments from this reviewer. Actually, we showed a strong connection between AP-1 and IL-18 by using Chromatin immunoprecipitation-qPCR assay in Figure 5H. “We continued to check the c-Jun binding site in the IL-18 promoter region. The results of the ChIP-qPCR assay showed that HMGB1-induced binding of c-Jun to IL-18 promoter… (Figure 5H).” (Line 378─381)

Q5. In Figure 6G, the result of tubulin need to be replaced with clearer blot.

Authors’ response: We appreciate the comments from this reviewer. We have accordingly modified and changed the Figure 6G.

Q6. This paper used IL-18mAb for IL-18 signal inhibition. Because it is difficult for antibody molecules to enter cells to play a role. The author needs to explain the relationship between HMGB1 and IL-18, and whether IL-18 plays a role in cells or outside cells.

Authors’ response: We appreciate the comments from this reviewer. We are very sorry for making reviewer misunderstanding the unclear manuscript. Our in vitro data showed a strong and significant result that we used IL-18 neutralizing antibody to confirm HMGB1-induced IL-18 secretion from myoblasts. We accordingly modified the information in the Discussion sections:

“Our in vitro results show that the mechanism of HMGB1 promotes IL-18 production in skeletal myoblasts and differentiated myotubes, and that mRNA levels of muscle atrophy-related markers were suppressed in myoblasts after neutralizing IL-18 secretion.” (Line 502─505)

Q7. In the animal model induced by Glycerol, the literature cited by the author was a model by Acrolein instead of Glycerol. Please explain why or cite a more appropriate literature. In addition, Glycerol mainly result in skeletal muscle injury. It is suggested that the author adopt a more appropriate animal model of skeletal muscle atrophy in the future.

Authors’ response: We appreciate the suggestions from this reviewer. We have accordingly changed GIMA into GIMI in the whole text and legends. We also changed a new reference for glycerol-induced muscle injury. We accordingly included the information in the Discussion and References sections:

 “It appears that inhibiting HMGB1 rescues skeletal muscle differentiation that is associated with IL-18 and improves muscle fiber morphology.” (Line 419─420)

“[39] Kawai H, Nishino H, Kusaka K, Naruo T, Tamaki Y, Iwasa M. Experimental glycerol myopathy: a histological study. Acta Neuropathol 80(2) (1990) 192-197.”

Q8. Please add a schematic diagram of possible signal mechanisms.

Authors’ response: We appreciate the comments from this reviewer. We have accordingly added the Figure 7 as a schematic diagram.

Reviewer 3 Report

The work described by Ho et al addresses how HMGB1 mediates skeletal muscle atrophy through IL18. The work is presented well and in a logical manner.

The review comments are:

1. The Supplementary file was not accessible. Kindly provide in a format that is compatible with all PC like Word or PDF.

2. Line 253: While HMGB1 is a Damage associated molecular pattern (DAMP) released during cell death, was is surprising to see no change in cell viability with HMGB1 knock down?

3. Fig 1 A, B - difficult to comprehend, please provide a better color contrast. 

4. The quantification for MURF2 in the WB for Fig 1C - G7 cells - please check (1.44 and 1.99). 

5. Fig 1G - Is it the expression of the proteins (MyHC, Desmin) that is affected (translationally) or is it the viability?

6. Please provide in the figure legend what statistical test was used for comparison - seems like Students t test. Please use ANOVA if more than 3 grps are present. 

7. Fig 2A - Did the authors expect not to see a change for  TNF and Il1b, the most well characterized pro inflammatory mediators -  in G7 cells in response to HMGB1 treatment?

8.  In line with the observations in Fig 2L, do the authors suggest the existence of any alternate pathways for IL18 production other than that mediated by HMGB1? Please include in discussion. 

9. The time course experiments nicely show the mechanistic action - but have the authors observed a time dependent change in expression of RAGE? 

10. Fig 4D - is a not a very convincing blot. 

11. Fig 6 - The authors say n=10 for the experiments - but the graphs show only 3 per group. Why?

12. Please provide a better blot for 6G - b tubulin. 

13. Figs 6 I and J - do not reflect the images in 6H. 

14. Fig 6M - The fiber size is reduced but more in number in GIMA + shRNA. Any explanations?

15. Why have the authors focused only on RAGE - will the other receptors for HMGB1 like TLRs follow the same effect? Please include in discussion. 

Author Response

Point-by-point reply to the reviewers:

Reviewer 3:

Q1. The Supplementary file was not accessible. Kindly provide in a format that is compatible with all PC like Word or PDF.

Authors’ response: We appreciate the comments from this reviewer. We have uploaded all Supplementary data as a PDF file.

Q2. Line 253: While HMGB1 is a Damage associated molecular pattern (DAMP) released during cell death, was is surprising to see no change in cell viability with HMGB1 knock down?

Authors’ response: We appreciate the comments from this reviewer. Since the HMGB1 knockdown C2C12 was stable cloned by lentivirus system, knockdown efficiency is approximately 50% as shown by protein expression (Figure 1E).  Therefore, Hmgb1 knockdown had no significant effect on cell viability. We accordingly added the information in the Discussion sections:

“Based on our results of Figure 1E and 1J, Hmgb1 knockdown efficiency is approximately 50% which had no significant effect on cell viability, though HMGB1 is a damage associated molecular pattern (DAMP) which released during cell death.” (Line 458─461)

Q3. Fig 1 A, B - difficult to comprehend, please provide a better color contrast. 

Authors’ response: We appreciate the comments from this reviewer. We have accordingly modified and changed the Figure 1A and 1B.

Q4. The quantification for MURF2 in the WB for Fig 1C - G7 cells - please check (1.44 and 1.99). 

Authors’ response: We appreciate the comments from this reviewer. We have accordingly modified and changed the Figure 1C.

Q5. Fig 1G - Is it the expression of the proteins (MyHC, Desmin) that is affected (translationally) or is it the viability?

Authors’ response: We appreciate the comments from this reviewer. Since the HMGB1 knockdown C2C12 was stable cloned by lentivirus system, knockdown efficiency is approximately 50% as shown by protein expression (Figure 1E).  Therefore, the expression of MyHC, Desmin is translationally affected by HMGB1.

Q6. Please provide in the figure legend what statistical test was used for comparison - seems like Students t test. Please use ANOVA if more than 3 grps are present. 

Authors’ response: We appreciate the comments from this reviewer. We have accordingly used one-way ANOVA to re-calculate again and modified all legends. We have accordingly added the following information to the Materials and methods sections:

“Student’s t-test was used to compare the means between two groups. Statistical com-parisons of more than three groups were performed using one-way analysis of vari-ance (ANOVA) with Bonferroni’s post hoc test, and two-way ANOVA or multi-way ANOVA were used with more than two factors.” (Line 235─238)

Q7. Fig 2A - Did the authors expect not to see a change for TNF and Il1b, the most well characterized pro inflammatory mediators -  in G7 cells in response to HMGB1 treatment?

Authors’ response: We appreciate the constructive comments from this reviewer. Our in vitro results have finished more than 6 independent experiments (n≧6). HMGB1-induced the expression of TNF-α and IL-1β had different phenotype results in C2C12 and G7 cell lines due to a great disparity.

Q8. In line with the observations in Fig 2L, do the authors suggest the existence of any alternate pathways for IL18 production other than that mediated by HMGB1? Please include in discussion. 

Authors’ response: We appreciate the constructive comments from this reviewer. We have accordingly added the following information to the Discussion sections:

“Our in vitro results show that the mechanism of HMGB1 promotes IL-18 production in skeletal myoblasts and differentiated myotubes, and that mRNA levels of muscle synthesis-related markers were suppressed in myoblasts after neutralizing IL-18 secretion. These results suggest that inhibiting HMGB1 rescues muscle differentiation via an IL-18-dependent mechanism. However, since we couldn’t exclude the effects of other pro-inflammatory cytokines on muscle differentiation (Figure 2A), further studies are needed to explain this assumption.” (Line 504─510)

Q9. The time course experiments nicely show the mechanistic action - but have the authors observed a time dependent change in expression of RAGE? 

Authors’ response: We appreciate the comments from this reviewer. We have dose-dependent results of RAGE in both growth and differentiated medium in Supplementary Figure S2.

Q10. Fig 4D - is a not a very convincing blot. 

Authors’ response: We appreciate the comments from this reviewer. We have accordingly modified and changed the Figure 4D.

Q11. Fig 6 - The authors say n=10 for the experiments - but the graphs show only 3 per group. Why?

Authors’ response: We appreciate the comments from this reviewer. The mice were randomly separated into three groups (control, GIMI, or GIMI with HMGB1 shRNA treatment) (n=10). Due to budget limitation, we picked up three mice for micro-CT analysis and others for IHC staining and western blotting.

Q12. Please provide a better blot for 6G - b tubulin.

Authors’ response: We appreciate the comments from this reviewer. We have accordingly modified and changed the Figure 6G.

Q13. Figs 6 I and J - do not reflect the images in 6H. 

Authors’ response: We appreciate the comments from this reviewer. We have accordingly modified and changed the Figure 6H.

Q14. Fig 6M - The fiber size is reduced but more in number in GIMA + shRNA. Any explanations? Authors’ response: We appreciate the comments from this reviewer. We used glycerol to induce muscle injury for 9 Days. Since the effects of muscle regeneration are not complete (at least for 14 days), the fiber size is under injury like Figure 6M. It is consistent with the previous study (Chen et al., J. Cachexia Sarcopenia Muscle 2019; 10: 165–176)

Q15. Why have the authors focused only on RAGE - will the other receptors for HMGB1 like TLRs follow the same effect? Please include in discussion. 

Authors’ response: We appreciate the comments from this reviewer. We have accordingly added the information in the Discussion section:

“In our study, immunoblotting demonstrates the effectiveness of RAGE or TLR4 following HMGB1 stimulation in C2C12 cells (Supplementary Figure S2). RAGE and TLR4 are both candidates for HMGB1 receptors via IL-18-dependent mechanisms in muscle differentiation markers, but more research is needed to clarify the association between HMGB1 and TLR4 in skeletal muscle atrophy.” (Line 489─493)

Round 2

Reviewer 1 Report

Thank you for providing responses to my concerns and suggestions. The paper is certainly improved for making these changes. There still remains some small tweaks to the English language that may improve the readbaility of the paper, but these may be sufficiently dealt with during the proffing and editing stage. I would suggest you do pay close attention the the writing in order to effectively present the work in its best light.

Author Response

Authors’ response: We appreciate the suggestion from this reviewer. We corrected the writing by native English language usage edited again.

Reviewer 2 Report

1. Myoblast is mainly responsible for proliferation in skeletal muscle, and skeletal muscle atrophy is mainly related to the imbalance of protein synthesis and degradation of myotube. Please give a logic and rational explain in introduction or discussion why detecting the atrophic genes atrogin-1, MuRF-2, and myostatin in myoblast.

2. In Figure 1H, I, K, because MyHC is specifically expressed in myotube, the author needs to determine whether the myoblast of the the ordinate description is accurate.

3. The ordinate of Figure 5h should be level, not expression.

4. In the schematic diagram, muscular atrophy should be the result, rather than the cause of myoblast differentiation abnormal. In addition, the authors need to separately describe the effect of HMGB1 on the myoblast and myotube.

Author Response

Reviewer 2:

Q1. Myoblast is mainly responsible for proliferation in skeletal muscle, and skeletal muscle atrophy is mainly related to the imbalance of protein synthesis and degradation of myotube. Please give a logic and rational explain in introduction or discussion why detecting the atrophic genes atrogin-1, MuRF-2, and myostatin in myoblast.

Authors’ response: We appreciate the suggestion from this reviewer. We have accordingly added the information in the Discussion section:

“A previous study showed that removal of MuRF-1 reduces the development of skeletal muscular atrophy by 36% [57]. Another study showed mRNA levels of MuRF-2 was higher than MuRF-1 in response to 3-day differentiation [58]. Wai et al. reported that MuRF, atrogin-1, and myostatin are important muscle catabolic signals in muscle wasting [8]. Thus, our results showed that HMGB1 increased the expression of skeletal muscle atrophy markers (atrogin-1, MuRF-2, and myostatin) via IL-18 expression.” (Line 457─463)

Q2. In Figure 1H, I, K, because MyHC is specifically expressed in myotube, the author needs to determine whether the myoblast of the the ordinate description is accurate.

Authors’ response: We appreciate the suggestion from this reviewer. We have accordingly modified the Figure 1H, 1I, and 1K and changed the ordinates into “MyHC positive area (μm2)”, “Desmin positive area (μm2)”, and “cell numbers / field”. We have accordingly added the information in the Discussion section:

“A previous study revealed that a transition in MyHC isoforms occurs during myogen-esis and muscle cell differentiation [59]. Our results showed HMGB1-regulated MyHC expression in myoblasts and differentiated myotube might associate with myogenesis and muscle differentiation and require to be further investigated in the future.” (Line 463─467)

Q3. The ordinate of Figure 5h should be level, not expression.

Authors’ response: We appreciate the suggestion from this reviewer. We have accordingly modified the Figure 5H and changed the ordinates into “Relative IL-18 levels / input (fold of control)”.

Q4. In the schematic diagram, muscular atrophy should be the result, rather than the cause of myoblast differentiation abnormal. In addition, the authors need to separately describe the effect of HMGB1 on the myoblast and myotube.

Authors’ response: We appreciate the suggestion from this reviewer. We have accordingly modified the Figure 7.